The landscape of fear conceptual framework: definition and review of current applications and misuses

Bleicher Sonny S. bleicher.s.s@gmail.com
Ecology and Evolutionary Biology, University of Arizona , Tucson , AZ , United States of America
Tumamoc People and Habitat, Tumamoc Desert Research Laboratory, University of Arizona , United States of America
Roper James
Electronic publication date: 2017 Sep 12
Publication date: 2017
Volume: 5
Electronic Location ID: e3772
Received 2017 Feb 27; Accepted 2017 Aug 16
Copyright: ©2017 Bleicher
Copyright year: 2017
Copyright holder: Bleicher
License: This is an open access article distributed under the terms of the Creative Commons Attribution License, which permits unrestricted use, distribution, reproduction and adaptation in any medium and for any purpose provided that it is properly attributed. For attribution, the original author(s), title, publication source (PeerJ) and either DOI or URL of the article must be cited.
License URL: https://creativecommons.org/licenses/by/4.0/

Keywords: Animal behavior, Habitat selection, Yellowstone, Giving-up density (GUD), Spatial ecology, Evolutionary dynamics, Food-webs, Mechanisms of coexistence, Conservation, Wildlife management

Funding: The author received no funding for this work.

==============================
Landscapes of Fear (LOF), the spatially explicit distribution of perceived predation risk as seen by a population, is increasingly cited in ecological literature and has become a frequently used “buzz-word”. With the increase in popularity, it became necessary to clarify the definition for the term, suggest boundaries and propose a common framework for its use. The LOF, as a progeny of the “ecology of fear” conceptual framework, defines fear as the strategic manifestation of the cost-benefit analysis of food and safety tradeoffs. In addition to direct predation risk, the LOF is affected by individuals’ energetic-state, inter- and intra-specific competition and is constrained by the evolutionary history of each species. Herein, based on current applications of the LOF conceptual framework, I suggest the future research in this framework will be directed towards: (1) finding applied management uses as a trait defining a population’s habitat-use and habitat-suitability; (2) studying multi-dimensional distribution of risk-assessment through time and space; (3) studying variability between individuals within a population; (4) measuring eco-neurological implications of risk as a feature of environmental heterogeneity and (5) expanding temporal and spatial scales of empirical studies.

Introduction

The study of community ecology has developed from a study of how species affect each other in terms of resource competition to the study of how that competition affected community structure over evolutionary time (Morris & Lundberg, 2011). In other words, how species’ resource-use efficiency impacts interspecific interactions on an evolutionary scale—resulting in present day community structures shaped by extinction and speciation events (Vincent & Brown, 2005). This historical shift can be traced back to the model that first tested top-down trophic cascades (e.g., Paine, 1963), and large collaborative and conceptual efforts to explain the dynamics driven by predator–prey interactions (e.g., Hassell, 1978; Murdoch & Oaten, 1975; Rosenzweig & MacArthur, 1963). These efforts continue today, and predominantly focus on the study of non-consumptive effects of predators on entire communities (Appendix S1) (e.g., Kotler, 1984; Sih, Englund & Wooster, 1998). Joel Brown colloquially referred to these non-consumptive effects as the “ecology of fear” (Brown, Laundré & Gurung, 1999).

While the ecology of fear continued to focus on the means by which community structure impacts specific behaviors, some choose to broaden the study to the ecosystem level (e.g., Madin, Madin & Booth, 2011). Such theses assess ecosystem health using the trophic cascades as the basis for a new theory of behavioral cascades reverberating down the food chain and affecting habitat selection of species along the chain.

John Laundré (2001) called the effect, behavioral-pattern, resulting from these trophic cascades the “Landscape of Fear” (LOF). The use of the LOF, as a concept, is gaining favor as more studies investigate spatial dynamics in the distribution of populations using a community centric lens. This review has two main objectives: (1) clearly define the LOF, while dispelling common misuses of the term, and (2) discuss how the current literature uses LOFs, suggesting future trajectories possible for this growing research program.

Review Method

For the purpose of this review, I conducted a literature search for manuscripts that use the term “landscape(s) of fear” as part of their title, or within their keywords and abstracts. The search was conducted on Google Scholar©, Wiley Interscience Online Library©, JSTOR© and Thompson Reuters’ Web of Science©. Every manuscript found in the search was examined and if it studied spatial distribution of predation risk it was included in the database. Every manuscript was mined for the following information: definition of the LOF(s), article type (review, opinion, or empirical), publication aim (based on journal guidelines), method used and study system (in studies providing data), and the theoretical contributions each brings to the field (if any). Three manuscripts were added to the database (despite not mentioning LOF per-se in their abstracts, or being peer reviewed): my own PhD thesis (Bleicher, 2014) and two manuscripts that were cited regularly in other manuscripts (Zanette & Jenkins, 2000; Ripple & Beschta, 2003).

Defining the LOF

Among growing interpretation of the LOF concept it is critically important to provide a concise and clear definition. The LOF is a behavioral trait of an individual animal, and more commonly used on the population level. The LOF provides a spatially dependent, yet geographically independent, measure of the way an animal “sees” its world—its umwelt (cf. Uexküll, 1909). In other words, it is a short term measure of the way the animal perceives its environment based on the cost-benefit analysis of the trade-off of food and safety associated with foraging in specific areas of the habitat available to it (cf. Brown, Laundré & Gurung, 1999). As such, LOFs are affected by a large variety of biological, evolutionary and yes—sometimes geographic variables.

Predation risk

The most studied factor impacting an animal’s LOF is direct and perceived predation risk (cf. Laundré, Hernández & Altendorf, 2001; Laundré, Hernandez & Ripple, 2010). Within predation risk, three major factors impact the LOF: (1) diversity of the predator community, (2) predation intensity [activity of predators] and (3) information [how well can the animal predict the likelihood of being attacked] (Brown, 1999).

A forager has to strategically decide where to forage based on the type of risk presented by the predators it is likely to face. For example, the decisions a forager must take to manage risk from ambush predators will differ greatly from those it will take to manage risk from a flying predator. In a case of Negev Desert gerbils facing barn owls and vipers, gerbils showed an ability to alter their LOF to adjust to the owl, a larger perceived threat than vipers, during the nights an owl is present in the experimental vivarium (Bleicher, 2014). Similarly, the features of the LOF will change based on the predator activity levels. A number of studies use landscape rugosity, how “wrinkled” the landscape appears, as a means to express heterogeneity in patterns of perceived-risk distribution (Laundré, Hernandez & Ripple, 2010; Bleicher, 2014). The best variable proposed to measure this feature of the LOF is the mean rate of change in foraging tenacity over space (mean harvested resources/meter). The greater the risk, the steeper the difference between safe and risky zones in the LOF. Thus, mean rugosity should increase as predator activity levels increase (Brown & Kotler, 2004).

Energetic-state

Gallagher et al. (2017) offer the opinion that the field of LOFs and the field of energy landscapes (using energetic-expenditure to explain movement through space) should be combined as they are two facets of the same coin. In many ways, they are correct, however the field of LOF has already focused a large portion of its scientific effort to quantifying the LOF using energetic tradeoffs, and foraging in particular.

This was first justified using an example with cape ground squirrels (Xerus inauris) where the costs associated with the distance they must venture from shelter altered their perception of risk (Van Der Merwe & Brown, 2008). Distance from refuge was an exercise in adding non-direct predation costs into the calculation of the LOF. This is important since over short periods, and small spatial scales, metabolic costs and missed opportunity costs are constant across the landscape. Examining these costs thus allows the extrapolation of the LOF from the temporally and spatially much more sensitive predation cost of foraging.

The possible variables that could alter an animal’s LOF include both physiological and external variables. Individuals (and populations) should take greater risk based on the increase in stress imposed by drought, blight, disease and parasites. Assuming optimal foraging, perceived resource availability should affect forager decisions. Stressed foragers will likely visit patches of greater risk-variability (high likelihood of a patch not yielding resources) (Real & Caraco, 1986). Alternately, resource shortage, will drive the stressed foragers to take risks by moving greater distances in search of isolated high quality patches, as in the case of Simpson Desert dunnarts (Haythornthwaite & Dickman, 2006; Bleicher & Dickman, 2016). Another factor impacting the energetic balance of the LOF is parasite load, both physiologically and behaviorally (Raveh et al., 2011).

Seasonal variability brings with it resource shortages that can shift the risk-taking behavior, namely water shortage (e.g., Shrader et al., 2008; Tadesse & Kotler, 2011; Arias-Del Razo et al., 2012). In times of drought, thirsty herds of African savannah ungulates are known to descend to water holes teeming with crocodiles and other predators. In this case the probability of escaping the predators, though meager, is still lower than the probability of dying from dehydration. It is the balance of risk and energetics that governs the LOF choices in the majority of cases.

Demographics

The success of an individual, its fitness, is measured not by the amount of energy (food) it successfully harvests from the environment, but in the successful conversion of that energy into viable offspring. Thus, competition for mates and care for offspring have an important role in determining both resource needs and risk-taking probability in opposite direction. Because of the time sensitivity of both offspring care and mating seasons, these shifts in behaviors will temporarily change the LOF for each individual, and collectively for the population.

In a study of collared peccaries, mothers protecting offspring remained in the safety of a wadi while males, not concerned with offspring safety, were observed foraging on resources near a hiking trail frequented by 1,500 visitors daily (S Bleicher & M Rosenzweig, pers. comm., 2016). This phenomenon of parental intimidation and its deleterious reproductive consequence were observed in a study on song sparrows (Zanette et al., 2011). On the flip side, during courtship, risk-taking in males, as in examples of lek behaviors, can lead to increased reproductive success (Boyko et al., 2004).

Density dependence—intraspecific competition

Living in a group provides safety in numbers (Rosenzweig, Abramsky & Subach, 1997), however the intraspecific competition can result in deleterious impacts on the foraging efficiency of individuals affecting the entire population’s fitness (Berger-Tal et al., 2015). Spatially, ecologists assume ideal free distribution, suggesting that the populations will disperse when the environmental conditions do not meet ideal fitness returns for individuals (Morris, 2003). By changing the spatial resolution on which we make our observations, we undoubtedly will be exposed to different stories. On fine-grained resolution we can observe the decision-making process impacting individual, however on a larger, course-grained resolution we are generally privy to the dynamics of the entire population (Druce et al., 2009). No empirical manuscript has yet to test the effects of density dependence on the landscape of fear as they materialize for populations of different sizes. It is likely that this specific factor will become the focus of more studies in the future.

Community structure—interspecific competition

The study of habitat selection has been largely dominated by community studies testing strong-weak competitor pairs of species (e.g., Rosenzweig, 1973; Dickman, 1986; Abramsky, Rosenzweig & Subach, 2001; McGill et al., 2006). Most of these studies suggest that a strong, more aggressive competitors, forces the weak to forage in less profitable habitat (greater risk/lower resources). This can also manifest itself in temporal partitioning (Kotler et al., 2002). This evidence strongly suggests that the LOF of a population in isolation will not compare with the same population’s LOF when competing for resources.

A couple of studies thus far have identified (or referred to) the impacts of competition on the LOF of competing species. (1) Competition for resources had greater impact on habitat selection in lemmings. This was attributed to the fact that the foraging season is very short and risk aversion may lead to starvation over the arctic winter (Dupuch, Morris & Halliday, 2013). In another example, the competition for resources can manipulate the distribution of predators. (2) In studies at various sites in Australia, dingo presence suppressed mesopredator populations (Ritchie & Johnson, 2009). Similarly, the competition between the invasive mesopredators suppresses the population of the competitor. When foxes are hunted, the feral cat populations explode and vice versa (Glen & Dickman, 2005; Allen, Allen & Leung, 2015), this in turn has a trickle-down effect on prey species’ LOFs.

Evolutionary history—ghosts of predator and competitor past

Behavioral and applied ecologists rarely study the evolutionary history of their species. The conditions (environment, community structure, resources) in which the species evolved will determine the tools which the species has to assess risk. In turn, these tools are applied by individuals to make the strategic decisions that manifest themselves in their LOFs. Comparing populations that use different habitat, or convergent species from similar environment may provide insight to the role of a larger meta-scale, both temporal (evolutionary and seasonal) and geographic (continents), in determining each population’s LOF.

The only example I found for such a study, was a macroevolutionary study where populations of convergent desert rodents were brought to a common arena and exposed to predator present in both systems, vipers and barn owls (Bleicher, 2014). Rodents of the Mojave Desert, that evolved with vipers that have heat sensing capabilities focused on the snakes as the focal driver of their LOF. The owl presence only elevated the risk in the entire landscape.

Rodents of the Negev Desert, who evolved with snakes blind in the dark, fear owls above snakes (Kotler et al., 2016). As a result, they redraw their LOF based on the greatest risk in the environment. On nights with vipers alone, they identify the ambush sites of snakes and avoid those. On nights with both vipers and owls they avoid the flight paths of the owls in the arena (vivarium).

Misinterpretation—avoiding misuse

There are three major misuses commonly published in the LOF literature. It is important to state them and discuss how these could be easily avoided for the benefit of this research program.

(1) Using the LOF concept interchangeably with habitat use. Animals will avoid habitat they perceive as risky, therefore the LOF plays a critical role in habitat use and habitat selection. However, it is important to re-emphasize that it is the habitat quality that is responsible in shaping the LOF. Thus, using the LOF to explain habitat use needs to be approached with caution. While activity data can reveal habitat use, they do not necessarily reveal the LOF. As an example, a rich food source can attract foragers and enhance activity to a given area without reduced risk necessarily being involved. This confusion is perpetuated by the fact the 81% (63) of manuscripts apply the LOF as a descriptor for habitat-selection. This problem is exacerbated by the fact the majority of manuscripts that misinterpret the term are able to draw sound conclusions about habitat selection (e.g., Creel et al., 2005; Madin, Madin & Booth, 2011). I suggest here, that when using the LOF as a descriptor for habitat-selection, one should also cross reference other critical factors, such as availability of food, shelter (nest sites, borrows, cavities), water etc.

(2) Suggesting predators impose a LOF. It is very metaphorically colorful to suggest that the presence of a predator imposes a LOF (e.g., Massey, Cubaynes & Coulson, 2013), however, every population interprets risk cues even in the absence of predators. This misinterpretation is most commonly published in applied journals, with 47% (9) of the applied-ecology manuscripts making this type of statement. The appropriate expression of these ideas, must refer to changes in the perceived risk associated with features in the landscape. This distinction was best shown in a study of vervet monkeys responding to playback of alarm calls corresponding to different predator types. This experiment generated three dimensional LOFs based on elevation in trees and spatial-distribution of safe zones within the troop’s home-range (Willems & Hill, 2009). Each call-type changed the monkeys’ preferred habitat.

(3) Using the LOF as jargon without defining the term. Though not common, 4% (3) of the manuscripts used the term, LOF, without defining it.

General Review Results

Since the year 2000, 77 manuscripts (Appendix S2) were published using the term LOF (in title, keywords or abstract). The publication rate has been increasing steadily since 2001, with a mean of 5.1 ± 0.7 SE manuscripts per year (Fig. 1).

Figure 1 Cumulative number of manuscripts using the “landscape of fear” (LOF) as a significant descriptor of the study in the title, abstract or keywords.

The Buzz-Word category is a classification of manuscripts that defined the LOF in a way that differed from a spatial distribution of a populations’ behavioral response to the perceived balance of resources and risk of predation. *Only manuscripts published between January–April 2017.

Of these 77 manuscripts 75% are empirical tests that employ the concept, while the rest of the manuscripts discuss implications in form of review and opinion papers. The majority of papers (76%) were published in journals dedicated to general ecology and zoology (e.g., Ecology©, Oikos© and the Canadian Journal of Zoology©) (Table 1).

Table 1 Summary table for all published manuscripts using the term “landscape of fear” (in title, abstract or keywords), and distinction of manuscripts misinterpreting the term as “Buzz-Word” manuscripts.

		Total manuscripts	% Buzz-Word (# of publications)	
A. Manuscripts using the LOF concept:	
		78	26.9% (21)	
B. Published in a journal covering:	
	General Ecology	44a	14% (6)	
	Zoology	15a	7% (1)	
	Animal Behavior	9a	22% (2)	
	Applied Ecology/Wildlife Management	9	45% (4)	
	General Biology	8	75% (6)	
	Evolution	3	67% (2)	
C. Manuscript type:	
	Opinion	8	25% (2)	
	Review	12	25% (3)	
	Empirical	58	28% (16)	
D. Manuscript defines the LOF as a:	
	Landscape trait	23b	31% (7)	
	Individuals’ trait	15b	33% (5)	
	Populations’ trait	49b	12% (6)	
Notes.

a Some manuscripts are counted more than one time if journals cover a variety of fields (e.g., Journal of Animal Ecology is categorized both as general ecology and zoology.

b Some manuscripts have conflicting definitions or apply the LOF to describe a characteristic of multiple levels and are thus counted more than one time.

With the rise in popularity of the term, the rate of misuse has also increased significantly. Between 2001–2009 a mean of 11% of the publications (per annum) used a definition different than the one intended by Laundré and Brown based in the “ecology of fear” (Brown, Laundré & Gurung, 1999). Between 2009–2016 the rate of misuse of the term increased to a mean of 35% (Fig. 1). For the purpose of the discussion ensuing, I categorized manuscripts that use a definition other than “a variation on a behavioral descriptor of the perception of risk a population sense in the environment” as “buzzword” papers. A number of manuscripts used the LOF to describe features of the environment as belonging to an animal’s LOF or define LOFs as traits of an individual (Table 1). If the manuscripts referred to the LOF as an intrinsic perception of the way an organism balances risk and energetic gains they were classified as relevant (Table 1).

38% (31) of the manuscripts discuss theoretical implications of LOFs. Of these, about half make suggestions that are of particular mention (Table 2). These contributions included ways to describe LOFs’ features, novel applications for which the LOF framework, novel methods to measure the LOF, or discussions on the merit of LOF as a research group (see ‘Applications’ section).

Table 2 Theoretical development of the LOF as a research program.

Year	Manuscript	Major theoretical contribution	
2000a	(Jacob & Brown, 2000)	• The LOF combines both spatial and temporal assessments of risk	
	(Zanette & Jenkins, 2000)	• The LOF is a measure of distribution of stress within a physical landscape based on habitat quality	
2001	(Laundré, Hernández & Altendorf, 2001)	• Defining the LOF framework as the impact of relative danger in shaping prey behavior and habitat selection.	
2004	(Brown & Kotler, 2004)	• LOF changed based on levels of risk: predator community or predator activity levels.	
	(Ripple & Beschta, 2004)	• Linking food webs to the ecology of fear through examples where fear of wolves trickled down to increase in vegetation diversity (and where it did not).	
2007	(Kauffman et al., 2007)	• Predators tap into prey LOF in hunting site selection.	
	(Rypstra et al., 2007)	• The individual effect: intra-species completion and cannibalism affect the populations LOF.	
	(Heithaus et al., 2007)	• Behavioral state: health of individual affects its LOF.	
2008	(Van Der Merwe & Brown, 2008)	• The LOF as a cost benefit analysis of energy; measuring a LOF in kJ.	
2009	(Druce et al., 2009)	• Defining spatial and temporal scales as drivers of change in LOFs	
	(Ritchie & Johnson, 2009)	• Studying inter-guild competition using the LOF framework (apex-mesopredators)	
	(Willems & Hill, 2009)	• Information based LOF’s signals for specific predators.	
2011	(Matassa & Trussell, 2011)	• Using survivorship as a measure of non-consumptive predator effects on both spatial and temporal scales.	
2013	(Dupuch, Morris & Halliday, 2013)	• Using the LOF as a tools to compare competition pressures and predation risk.	
2014	(Bleicher, 2014)	• Defining LOF shape and plasticity; The LOF as a tool for macroevolutionary comparison.	
2015	(Hammerschlag et al., 2015)	• Linking activity patterns of predators the LOF of prey on a temporal scale.	
2017	(Laundré et al., 2017)	• Comparing bottom-up and top-down models of population dynamics using the LOF framework.	
	(Gallagher et al., 2017)	• Combining LOF and energy landscapes as one unit.	
Notes.

a These references to the LOF were published prior to the seminal paper Laundré, Hernández & Altendorf (2001), however are regularly cited as influential papers in the field, or had referenced the seminal paper as unpublished work.

Measuring the LOF

From this point in the manuscript I will only refer to a subset of the manuscripts that used my definition of a LOF (57 in total). The majority of these manuscripts focused on ungulates in North American alpine scrubland systems (Tables 3A and 3B). It was the wolf-elk-willow system that brought the LOF into common ecological jargon through the study of successful reintroduction of wolves to Yellowstone National Park. Despite the base of the LOF in ungulate research, many studies preferred the variability provided by small-mammal model species (gerbils, heteromyids, lemmings and voles). Researchers have manipulation capabilities in small mammals resulting in 40% (6) of these studies being performed in controlled captive environments (vivaria). These vivaria allow for the manipulation of predation risk and environmental conditions: e.g., homogenous landscapes, illumination, resource availability, energetic state of the population etc. The rise in small-mammal experiments also secured the giving-up density (GUD; cf. Brown, 1988) as the preferred measure of LOFs (Table 3B).

Table 3 Summary table for landscape of fear studies, empirical and opinion manuscripts, which defined the LOF as a behavioral trait of the studied population.

(A) Classification by system type and continent. (B) Classification by measurement of fear and study focal organism.

(A)		Continent	
		N. America	Africa/Sahara	Australia	Europe	Asia/Polynesia	Total	
Study system	Alpine scrubland	9	4				13	
	Arid/Tundra	4	8				12	
	Temperate forest	1		2	3		6	
	Grassland/Savannah	1	3		1		5	
	Marine	2		1		1	4	
	Anthropocentric				1	1	2	
	Total	17	15	3	5	2	42	
(B)		Focal Organism	
		Ungulate	Small mammal	Predatorsa	Primates	Marine herbivoresb	Vegetationc	Birds	Total	
Measurement variable	GUD	6	13	1	4	1	1		26	
	Occupancy	2	1	2		1			6	
	Scat density	2	1						3	
	Telemetry	1		1		1			3	
	Vigilance	3							3	
	Others	1					1	1	3	
	Total	15	15	4	4	3	2	1	44	
Notes.

a Both mesopredators and apex predators.

b Multiple studies used grazing reef fish as a group as opposed with a specific species.

c Damage to algae or woody vegetation.

Given species-specific constraints, each class of organisms has a specialized-tool kit used to measure its LOF (Table 3B). GUDs remain the most versatile measure for the studies using habitat assessment (59% of the manuscripts) and historically were successfully used to measure environmental stress in birds, ungulates, small mammals and experimentally—fish (Bedoya-Perez et al., 2013). Larger mammals and finicky foragers pose a challenge to the GUD method. As a result, the LOFs for larger species were commonly measured applying occupancy models using variables such as scat abundance, hoof mark density and trail camera arrays (19% of manuscripts). Measuring predator (and marine) LOF provides even further challenge due to low density of populations. Thus, the major tool used was radio and GPS tracking. Despite being a very small proportion of the current literature base, some efforts have been made to quantify environmental risk using stress hormones. So far, this method has been limited to birds (Chalfoun & Martin, 2009; Roper, Sullivan & Ricklefs, 2010; Clinchy et al., 2011).

Thus far, 38% (17) manuscripts provided a map for the study organism’s LOF. These maps help readers relate with associated distribution of risk the studied population experiences. The majority of these have used the GUD as the measure of risk and graphed the LOF map as a three dimensional scatter plot, using a distance-weighted-least-squares (DWLS) smoothing function to generate the contour lines (or raster) for the maps (Fig. 2).

Figure 2 Example of Landscape of Fear Map using a dataset adapted from Bleicher et al. (2016).

The map shows the distribution of risk using giving up densities (GUDs) for a population of Allenby’s gerbils (G. andersoni allenbyi) in a controlled enclosure in Sde Boker, Israel. The contour lines are derived using the distance weighted least squares (DWLS) smoothing function at a tension of 0.5. GUD values above 2.0g (orange and red) reflect areas that are perceived as dangerous by the gerbils while areas below 1.0g (green and blue) reflect zones of safety. The + signs are the locations where data were collected and both x and y-exes are measuring the enclosure in meters. This figure was generated using Systate13®.

The current literature linguistically borrows attributes from other ecological, evolutionary and geographic theories to describe the zones of different risk characteristics. For example, in a study of striped mice, the features of risky habitat was described as “islands” of fear, a reference to the island-biogeography theory and the SLOSS debate (cf. Diamond, 1975). This study emphasized the impacts of both borders and edges in altering the distribution of safety zones the mice perceive in the environment (Abu Baker & Brown, 2010).

In a previous review, Laundré, Hernandez & Ripple (2010) prefer to describe the landscape features as valleys and peaks (re: safe to risky) in an aim to show that risk assessment is a quantitative attribute and not a binomial characteristic (two distinct outcomes risk or safety). Lastly, my own work evaluated the role played by gradients of change (in perceived risk) describe the attributes of the decision making process made within the LOFs.

This measure can be described as the rugosity of the landscape (Bleicher, Kotler & Brown, 2012; Bleicher, 2014). A highly rugose landscape (highly variable with steep changes between points) implies that the population perceives the risk as localized. In comparison, flat landscapes can be interpreted as the result of one of two distinct behavioral strategies. (1) A flat LOF may be the result of a very “fearful” population whereas the majority of the environment “plateaus” on a high risk contour. In such a LOF, the major focus of the behavior remains in contact with the locations of refuge in the landscape and the risk lessens gradually as one moves near the refuge. Alternately, (2) a population that is “secure” in its management ability of predation risk from the predators in the environment will have a very flat (low) landscape. In this scenario, the zones of risk are less focused and tangible and thus the change between “riskier” and “safer” zones is gradual and not very distinct.

Current applications

Of the manuscripts defining a LOF as a trait of population behavior, 42 were empirical. Within those it is possible to divide the aims of the manuscripts into four focal aims: (1) to characterize the role of perceived predation risk on habitat use by wild and captive populations, (2) to project top-down trophic effects, (3) to understand how habitat complexity affects demographic and behavioral dynamics within populations and (4) to deconstruct community interactions within an evolutionary framework.

Population level

The majority, 76% (33), of empirical manuscripts using the LOF, and almost all, 92%, of the manuscripts that provided visual charts of the LOF, aimed to study how populations perceive their environment. Thus, the majority of publications apply the LOF as an equivalent to habitat selection. As mentioned above, this has given rise to a large misinterpretation of the LOF. The large number of publications here makes the review of these largely unnecessary. However, there are a couple of noteworthy examples that did impact the understanding of the LOF.

Druce et al. (2009) showed in his study of klipspringers that the study-scale can reveal different patterns of elements impacting a LOF. In this study, the small scale (grids of stations 3–4 m apart) showed that microhabitat (cover, distance from rocky outcrops) impacted forgiving decisions specifically on a temporal scale. But on larger scales (grids of 6–24 stations 30 m apart), the major geographic features of the landscape (substrate, drainage lines) explained the majority of variation in foraging decisions. This study drew the attention to the importance of natural history to calibrate experiments to study the focal population on terms relevant to their specific ecology.

Kauffman et al. (2007) showed that the conspecific competition had a stronger impact on the distribution of wolf kill-sites than habitat suitability for hunting. This study made two noteworthy contributions to the understanding of fear-based habitat use. (1) Predators are constrained in their activity by elements beyond prey availability and ease of hunting. (2) Information about predator limiting factors, gained through experience cohabitating with the predators, will alter prey decision-making. In this case, given enough information, prey will likely prefer habitat of territorial dispute between wolf-packs.

Trophic dynamics

Despite the fact that the origin of the LOF framework is in trophic cascades, only half (22 total and 13 empirical) of the manuscripts actually include multi-trophic studies (Table 4). Noteworthy examples of multi-level studies include the trickle down of shark predators on algal blooms in tropical reefs (Madin, Madin & Booth, 2011). Additionally, Manning, Gordon & Ripple (2009) offer a predictive study of the impact reintroducing wolves to Scotland would have on vegetation patterns. This study drew direct parallels to vegetation regeneration post the reintroduction of wolves into Yellowstone National Park (Beschta & Ripple, 2009; Ripple & Beschta, 2012).

Table 4 Distribution of manuscripts by trophic levels studied.

	No. of publications	
A. Number of trophic levels in study	
1 Trophic level	20	
2 Trophic levels	23	
3 Trophic levels	6	
B. Focus of study	
Humans	1	
Apex-predators	2	
Carnivore	49	
Herbivore (granivore)	31	
Vegetation	10	
**Non Biotic	1	

The majority of the studies that did use a trophic framework, only looked at a pair of species (predator–prey) (Table 4). Interesting examples of these include the study of predator facilitation and interference (e.g., Ritchie & Johnson, 2009; Embar et al., 2014). Similarly, there is a fair number of studies that question the role of predation risk in the distribution of the prey populations. Two good studies can be shown for this category: (1) Coyote distribution (scat) does not correlate with the distribution of jackrabbits and other desert rodents in Chihuahuan high desert (Laundré, Calderas & Hernández, 2009), and (2) sharks and sea turtles do not show the same pattern of spatial and temporal rates of surfacing behavior (Hammerschlag et al., 2015).

Individual-based LOFs

This category of applications represents a very small proportion (5%) of the manuscripts, however that subset is of utmost importance in developing the study of LOFs. Those manuscripts define the LOF as a population trait, however they acknowledge individual variation within the population. Each of the three studies took a very different approach.

Zanette & Jenkins (2000) measured fledging success and correlated it with predator activity within fragmented forest segments. They suggest that the more fragmented the habitat, the more stressed the parenting birds are, and thus the offspring are less likely to fledge. On the flipside, this suggests that parental stress becomes a predictor of predation risk, or predator distribution.

Rypstra et al. (2007) found that wolf-spiders exposed to a larger predatory spider were driven into a mixed habitat where their prey capture rate was significantly diminished. Individuals who were not exposed to the larger spiders were found in exposed, i.e., risky, habitat and were found to have a greater hunting success. This study suggests that gaining information about predator preferences (through cues) causes a shift in an individual’s LOF. This approach provides insight into the learning process, or loss of naiveté, that is hard to observe in natural settings.

The third study was performed on state-dependent risk taking in green sea-turtles (Heithaus et al., 2007). This study found that turtles with low fat reserves were likely to forage in shark infested waters, while healthier individuals remained in shallow waters and low shark habitat. This study shows that individual well-being affects the way that individual perceives the tradeoff of food and safety. As a result if the LOF was measured for groups of turtles based on their energetic state, a different shape would be revealed.

Evolutionary mechanisms of coexistence

The LOF, as a derivative of the ecology of fear (Appendix S1), relies on Darwinisitic evolutionary forces to explain the ecological dynamics associated with communities, populations and individuals. This suggests that the forces historically influencing the study of populations’ ancestors, ghosts of predator and competitor past, would mold the way they present day descendants respond to the tradeoffs of food and safety.

Only two empirical studies applied an evolutionary lens to their discussion. The first found that lemmings that evolved in arctic conditions, with limited time to store resources for the long winters, give precedence to competition over the risk of predation (Dupuch, Morris & Halliday, 2013). This study provided the incentive to use the LOF to ask questions comparing species within the same trophic level. It provided the framework to measure competitor strategies using spatial distribution of risk perception. This study inspired the four way comparison (captive study) of convergent rodents from two continents under the predation risk of predators shared by both systems (Bleicher, 2014).

In this study, two heteromyid rodents from the Mojave Desert and two gerbil species from the Negev Desert were exposed to treatments of vipers and owls in a homogeneous semi-natural arena. The heteromyids that evolved alongside vipers that use heat-sensing pits to “see” in the dark exhibited fixed LOFs that did not change their shape when owls were added to the vipers constantly present in the environment. Meanwhile the middle-eastern gerbils, who evolved with snakes “blind” in the dark exhibited plastic LOFs. They altered their LOF, adjusting the topography in ways that address risk that is derived from unique adaptations to manage risk from different types of predators. In other words, focusing the peaks and valleys on the activity pattern exhibited by the predator they perceived as the greatest risk. For these gerbils, this meant the owl (Kotler et al., 2016). In its absence the LOF peaks were centered around viper ambush sites.

Prospectus—developing the LOF for future applications

I would like to “throw the gauntlet” to my colleagues and offer the following five directions in which the LOF concept can be applied.

Conservation and applied management

Despite the theoretical background of the LOF in conservation efforts and the reintroduction of wolves to Yellowstone National Park, the active management of populations has not measured LOFs as a monitoring tool. Charting the LOF can provide a temporal snapshot of the way populations see their environment. With relatively low effort (installing a food patch matrix), and in a short time-frame (4–15 repetitions), one can, for example, ascertain the efficiency of a habitat augmentation program (cf. S Bleicher & C Dickman, 2015, unpublished data). Similarly, the LOF can provide an accurate measure of the impact of human activity has on species of conservation concern without waiting for demographic changes in the population. Additionally, one could use the LOF to physically study how we can increase the perception of risk a pest population senses in an area (cf. S Bleicher & M Rosenzweig, 2016, unpublished data). By cues of predation risk (sound, odor) as management treatments (e.g., Suraci et al., 2016), one could follow the changes in spatial distribution of the pests. Such methods could be applied, for example, in air-fields to lower wildlife-impacts and control agricultural pests.

The LOF studied on meta-scales

The macro-evolutionary perspective—comparing the populations, species and communities that evolved under different conditions—requires comparisons on very large geographic scales as well as cross-species comparison. To achieve such comparisons studies have two possible designs: (a) run common-garden experiments bringing populations with varying evolutionary background to a shared location (e.g., Bleicher, 2014; Kotler et al., 2016; S Bleicher, B Kotler & J Brown, 2014, unpublished data) or (b) run comparative studies within areas of similar habitat at geographically isolated locations. Such studies have the ability to shed light on how predator–prey dynamics of the past drive current-day mechanisms of species coexistence.

Above, I suggested that the environmental conditions at the time data is collected to generate the LOF, impacts the shape of the LOF. As LOFs provide a short-term “picture” of how populations perceive their environment, field-studies will require repeated measures to understand how risk perception is affected by drought, population booms or busts, predator density or activity. Similarly repeated measures along an extended period of time would be required to monitor habitat restoration or augmentation. For example, a study that used the LOF to look at the effectiveness of shelter augmentation (S Bleicher & C Dickman, 2015, unpublished data) suggested that the study must be repeated once the populations recovered from the bust induced by bush-fires and a prolonged drought. As in this example, the shape of the LOF should alter based on the increased inter- and intra-specific competition and by the additional resources and shelter that successional vegetation will provide. Despite this well-known (and discussed) temporal constraint, no studies have yet to follow-up with repeated measures. The exception to this is the follow-up on the Yellowstone wolf reintroduction that looked predominantly at plant diversity and abundance of aspens and willows but did not measure the elk or bison’s LOF (Ripple & Beschta, 2012).

The 4D LOF

Mapping the LOF provides a level of intricacy that categorical analysis can fall short of explaining. The growing number of studies offering contour maps (3D scatterplots) of the LOF is a sign for the increasing prominence of spatial statistics in current ecology. Studies in primates and brushtailed possums suggest that elevation has as much significance as landscape heterogeneity in the management of risk from a variety of predators (Willems & Hill, 2009; Emerson, Brown & Linden, 2011; Mella, Banks & Mcarthur, 2014). Similar to those mammals, most species do not live on a two dimensional plane. Therefore, one must conclude that the future of the field will aim towards 4D and 5D models that incorporate altitude (aerial, aquatic or above/below ground) and time (hourly, seasonal, annual or generational).

The personality-based individual LOF

Populations are comprised of individuals with differing phenotypic expressions on an axis limited by the niche breadth (range of possible expression forms) of each trait (Vincent & Brown, 2005). When we measure variability on the population level we average out the “noise” produced by the variation in individual response. The focus on the individual is gaining prominence in behavioral ecology. Specifically the study of bold-shy expression and how they affect risk assessment (e.g., Ioannou, Payne & Krause, 2008) and the spread of invasive species (e.g., Fogarty, Cote & Sih, 2011) as a couple of examples.

Variability of traits within a population is a pillar of Darwinistic evolution, and I suggest that the LOF may provide a platform to study consequences of changing behavioral traits. Manipulative experiments could subject stressors on a selective group within a population, following how these treatments change the individual’s risk assessment from the overall population. Some of these manipulative studies already exist, however they did not focus on the spatial components of the behavioral response. An example of such a study is the effect of parasites and the distraction they produce on the risk-taking behavior in gerbils (e.g., Raveh et al., 2011).

Some manipulative studies of this sort could involve: group size and density dependence (does group size influence the boldness of individuals?); energetic-state (does the hungry individual take greater risks than satiated individuals in a group?); demographics (does a male take more risk when competing for mates with many other males?, or do females with offspring reduce the risk-taking in comparison with the males in the group?) and more.

The neurology of LOFs

The last frontier to the LOF studies I wish to highlight is the neuro-ecology of fear. i.e., converting environmental risk into a measureable impact on stress syndromes. When an animal is under stress (risk of predation specifically), the neurological registering of the risk cues causes an increase in stress hormones being released in the body of the animal (Gross & Canteras, 2012). The physiological responses to these stress hormones are energetically costly (Apfelbach et al., 2005) and result in lowered productivity (e.g., Mukherjee et al., 2014).

For example, sparrows are shown to respond with an increase of a variety of stress hormones (plasma total corticosterone, corticosteroid binding globulin (CBG) and free corticosterone) in response to an increase in the risk of predation in the environment (Zanette et al., 2011). Creel et al. (2013) suggest that competition may play a similar role in producing stress hormones, and should result in changes in population dynamics. I agree with Clinchy, Sheriff & Zanette (2013) in suggesting that this connection of environmental stress and neurological responses is a fertile ground for research. It is important to move away from the chronic stress studied in laboratory animals into spatially explicit studies within realistic ecological scenarios.

Supplemental Information

Appendix S1 APPENDIX I: the historical base leading to the LOFs

Click here for additional data file.

Appendix S2 APPENDIX II: bibliography of manuscripts reviewed for analysis

Click here for additional data file.

I would like to thank BP Kotler, JS Brown, CR Dickman, JW Laundré and ML Rosenzweig for conversation and guidance that inspired this research. I would like to extend a special thanks to A Halloway, J Bernhart, A Adivrekar and D & L Bleicher for editorial comments both in contents and in copyediting.

Additional Information and Declarations

Competing Interests

Author Contributions

Data Availability

The author declares there are no competing interests.

Sonny S. Bleicher conceived and designed the experiments, performed the experiments, analyzed the data, contributed reagents/materials/analysis tools, wrote the paper, prepared figures and/or tables, reviewed drafts of the paper.

The following information was supplied regarding data availability:

This is a review manuscript. The research in this article did not generate, collect or analyse any raw data or code.

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
