# Peer review of "The landscape of fear conceptual framework: definition and review of current applications and misuses"

_PeerJ, doi:10.7717/peerj.3772_

## Round 0.1 · original submission · Major Revisions

As a review of the current state of understanding of LOF, the reviewers had very different opinions on manuscript quality. In my own reading, I feel that perhaps it is somewhere between the two. I agreed with the first reviewer about the writing style and the literature review and how you commented on the literature. I would have liked to have seen sort of an evolutionary progression of how LOF has been addressed over time, whether some or many were mistaken conceptually, what their contributions were and so on, and how they may have converged on a coherent concept today. The three figures could be combined into one, with three panels - conceptually they are very simple.

The review by Laundre was much simpler, and in some ways, less useful as far as suggestions for improvement go. But, as he is an important player in this area, I respect his perspective.

Of course with a review paper, it is hard to touch on all topics, but as the first reviewer suggested, and I to a lesser extent in my revised version, some of the literature could be a bit better reviewed, with deeper explanations of their implications.

Finally, I also thought that, in places, the writing style was awkward. I did not, in this version, comment on every single instance where I thought the text could be improved, but I do provide several examples - please take those examples to heart and try to see the patterns they represent, and edit your text accordingly.

# Note from Staff: While revising, please pay attention to our guidelines relating to literature review standards (https://peerj.com/about/policies-and-procedures/#discipline-standards) and standard manuscript sections for literature reviews (https://peerj.com/about/author-instructions/#literature-review-sections) #.

Reviewer 1 ·

Basic reporting

The literature review is very weak. The author did not dig sufficiently deeply in many areas and omitted some important literature that would have been essential to understand the question posed. The author did not exercise a sufficiently high level of critical thinking by fully considering counter arguments (e.g., bottom-up factors and how they interact with top-down factors that include behavioral responses to avoid predation).

Experimental design

The design of the paper is very weak as are the methods. It is unclear whether experimental studies were conflated with non-experimental work for this meta-analysis.

Validity of the findings

Based on this reviewer's responses to items 1 and 2, the findings are not sufficiently valid to support being published as a peer reviewed journal article.

Additional comments

This analysis is not publishable as presented. The line-edit notes present some suggestions. But this needs to be re-done, approaching the topic in a far more critical manner. Suggest focusing only on empirical Landscape of Fear studies (ones that were done using experimentation, GPS collars, etc.) as the ones that do not are too subjective and have high potential for bias.

Annotated reviews are not available for download in order to protect the identity of reviewers who chose to remain anonymous.

·

Basic reporting

No Comment

Experimental design

This is a review paper so no comment.

Validity of the findings

No comment

Additional comments

Over all this is a good review of a fast growing and changing area of ecology, the landscape of fear. The author does a good job of reviewing three areas he wished to evaluate: History, methods, and current and possible future uses of the concept. I agree with the author that one of the most significant ways the LOF model can be applied is in the area of conservation. The ms should provide a good current "state of the art" assessment of where the LOF is and where it potentially can go in the future.

---

## Round 0.2 · Minor Revisions

I agree with the comments of the new reviewer and I appreciate the extent to which this reviewer makes comments on your manuscript. Please attend to the details and in your rebuttal letter, please explain how you resolve each item well. I agree that the original title was very ambitious and I am not sure the article lived up to the expectations of the title. I also note that the English style is still somewhat awkward and would like to see improvements there.

·

Basic reporting

No comment.

Experimental design

Not applicable.

Validity of the findings

No comment.

Additional comments

General Comments

As a review paper on the Landscape of Fear concept, this manuscript makes a valuable contribution to the field of behavioral ecology, but I am not in complete agreement with the definition (and especially its scope) as it is used in this paper (see below under Specific Comments).
Apart from minor typos and problems with grammar (see some examples below) the manuscript is well-organized and the cited literature is relevant and well-researched. Figures and tables are informative and support the text. I would like to see a more explicit discussion/explanation of the role of scale in Landscape of Fear studies (see more details below).
I am not sure that I am comfortable with the title of the manuscript. While the manuscript does review and clarify the Landscape of Fear conceptual framework, it does not really redefine it. Or if it is, I am not sure that I agree with the new definition. Something along the lines of “A review and definition of the Landscape of Fear concept: applications and misuses” may be more appropriate.

Specific Comments:

The English language of the manuscript still needs improvement. Some examples and suggestions from just the Introduction:
1) Lines 27: replace “with” with “in”
2) Line 29: “the birth of the predator-prey dynamics research group” requires a reference that either documents or defines the origin and goals of the group.
3) Lines 30-31: awkward sentence.
4) Line 32: presumably “dynamics” refer to the “non-consumptive effects” from the preceding sentence. Stick with “non-consumptive effects” for both sentences. It will make it easier to follow along while reading.
5) Line 35: replace “onto an” with “to the”?
6) Line 40: insert “the” in front of “distribution”
There are many more examples scattered throughout the rest of the manuscript. I am not a native English speaker and not the best person to fix typos, ambiguity, and grammar, but the manuscript will benefit from more careful editing.

Line 39: Which effect exactly? The cascades? But “cascades reverberating down the food chain” are processes, not effects. On the other hand, I do agree that the Landscape of Fear concept describes an effect or pattern, but it is not the trophic/behavioral cascades themselves. In other words, the current wording confuses ecological processes (the cascades) with patterns (the Landscapes of Fear).

Line 59: While the Landscape of Fear can be a population trait, I do not agree that the Landscape of Fear should be restricted to being a population trait. Landscapes of Fear are best described as short-term and small-scale patterns that result from foragers’ cost-benefit analyses of the trade-offs between food and safety. In principle, a separate Landscape of Fear exists for each individual forager. In practice, we often combine data collected from different animals to describe a Landscape of Fear for a local population over a defined period (which can then be viewed as a population level trait). However, as the author points out later in the paper, an exciting avenue for future research is using the Landscape of Fear concept in the context of state-dependent foraging. In this context, the Landscape of Fear is more likely to be investigated as a trait of the individual rather than the population. In other words, I do not agree with author’s definition of the Landscape of Fear concept, which restricts it to a population level trait.

Lines 57-170: I would like to see a more explicit discussion on the role of scale as it relates to the Landscape of Fear concept. At this stage, the theoretical foundation and field methodology for the Landscape of Fear concept are well-developed. In my humble opinion, the next big step forward for the Landscape of Fear paradigm will come from empirically supported meta-scale studies. Such a discussion will support and clarify the “Prospectus” section of the manuscript. For example, the Van der Merwe and Brown (2008) (line 94) study treated Landscapes of Fear as spatial patterns that result from food/safety trade-offs measured over short foraging periods and at relatively small spatial scales (at heart it is a methods paper). This is important since it is only over short periods and small spatial scales that metabolic costs and missed opportunity costs will be constant across the landscape, thus revealing the Landscape of Fear from the temporally and spatially much more sensitive predation cost of foraging. Such short-term landscapes are of course dynamic over longer temporal and spatial scales due to many of the factors described in lines 66-170. How many studies have explicitly looked at this meta-scale with data that are able to show the long-term and large-scale dynamics of Landscape of Fear patterns? When you talk of something like intraspecific competition (line 116) or drought (line 108) shaping Landscapes of Fear, these are factors that can only do so at this meta-scale level (in these cases via affecting the missed-opportunity cost of foraging, which is not going to vary within a local and short-term Landscape of Fear). Note that the Druce et al (2009) study, while mentioning scale, is more about the spatial resolution of data collected from fine-grained vs. coarse-grained foraging grids, rather than the LOF scale differences that I am talking about here. The author’s own dissertation work (Bleicher 2014) looks like an excellent example of the large/meta-scale that I am talking about, but I suspect that such meta-scale studies are still few and far between (though I am sitting behind some formidable scientific literature paywalls at my institution!)

Lines 175-181, and especially line 176: I agree that the LOF concept is not the same as habitat use and confusion in this regard needs to be avoided. However, the LOF plays a critical role in habitat use/selection and using LOFs to draw at least some conclusions about habitat selection is a valid approach. Can you provide examples of papers that generate a LOF that is then used as the sole descriptor of habitat use? More problematic in my mind is the opposite: studies that claim to generate LOFs from habitat use or activity data only (see lines 233-235 for examples). In other words, while activity data can reveal habitat use, they do not necessarily reveal the LOF (for example, a rich food patch can attract foragers and enhance activity, without reduced risk necessarily being involved). Perhaps this is what you meant all along with your own argument in this section? In which case clarifying the matter, with specific examples, will be the way to go.

Lines 182-191: Yes, LOFs are about perceived risk of injury (and strictly speaking not just from predators!) and do not result directly from the simple presence/absence of predators per se, but are some authors actually implying that predators impose LOFs regardless of physical features of the landscape? If so, can you provide specific examples and references of this?

Lines 357-419: These are all exciting new developments, but as discussed above, it is worth mentioning that we need more future focus on empirical studies that look at LOF dynamics at larger and longer scales.

---

## Round 0.3 · accepted · Accept

I appreciate your effort to improve the English and the clarity of writing style. There are still a few items that could be fixed - for example, you use the word "manifest" in the abstract and introduction that should be "manifestation" (at least sometimes). Also, you say "inter-species interactions" and I would think "interspecific interactions" would be more appropriate. However, the manuscript has improved over its several "manifestations" and so I have decided to accept it.